# Application of Piezoelectric PLLA Braided Cord as Wearable Sensor to Realize Monitoring System for Indoor Dogs with Less Physical or Mental Stress

**DOI:** 10.3390/mi14010143

**Published:** 2023-01-05

**Authors:** Yoshiro Tajitsu, Jun Takarada, Tokiya Hikichi, Ryoji Sugii, Kohei Takatani, Hiroki Yanagimoto, Riku Nakanishi, Seita Shiomi, Daiki Kitamoto, Takuo Nakiri, Osamu Takeuchi, Miki Deguchi, Takanori Muto, Kazuaki Kuroki, Wataru Amano, Ayaka Misumi, Mitsuru Takahashi, Kazuki Sugiyama, Akira Tanabe, Shiro Kamohara, Rei Nisho, Koji Takeshita

**Affiliations:** 1Electrical Engineering Department, Graduate School of Science and Engineering, Kansai University, Suita 5640-8680, Japan; 2Tokyo IoT Technology Department, 5G & IoT Engineering Division, Softbank Co., Kaigan, Tokyo 105-7529, Japan; 3Revoneo LLC, Fushimi, Kyoto 600-8086, Japan; 4Renesas Electronics Co., Ltd., Toyosu, Tokyo 135-0061, Japan; 5Teijin Frontier Co., Ltd., Kita, Osaka 530-8605, Japan

**Keywords:** poly-*l*-lactic acid, braided cord, piezoelectricity, wearable sensor, monitoring system

## Abstract

We attempted to realize a prototype system that monitors the living condition of indoor dogs without physical or mental burden by using a piezoelectric poly-*l*-lactic acid (PLLA) braided cord as a wearable sensor. First, to achieve flexibility and durability of the piezoelectric PLLA braided cord used as a sensor for indoor dogs, the process of manufacturing the piezoelectric PLLA fiber for the piezoelectric braided cord was studied in detail and improved to achieve the required performance. Piezoelectric PLLA braided cords were fabricated from the developed PLLA fibers, and the finite element method was used to realize an e-textile that can effectively function as a monitoring sensor. As a result, we realized an e-textile that feels similar to a high-grade textile and senses the complex movements of indoor dogs without the use of a complex computer system. Finally, a prototype system was constructed and applied to an actual indoor dog to demonstrate the usefulness of the e-textile as a sensor for indoor dog monitoring.

## 1. Introduction

COVID-19 pandemic triggered changes in the way people work, with many people spending more time at home. Amid these changes in the work and living environments, the number of households in Japan that keep pets increased by nearly 20%, with dogs now accounting for about 80% of all pets kept indoors [1]. The BBC reported that the number of new dog owners during lockdown in the UK was 3.2 million [2]. This trend can be seen in many countries. Newly bought puppies, known as “pandemic puppies” [3], have been living in close proximity to their owners since their acquisition. Thus, these puppies have become particularly distressed at being left alone and have developed problematic behavior [4,5]. As a result, there is a growing movement in the field of pet monitoring to solve this problem by using IoT devices with remote connectivity capabilities. In other words, surveillance-type pet monitors are now being installed. Although surveillance-type monitoring is very convenient, it is difficult to cover every room and site when the owner is not present. In addition, there are many nonvisible areas such as under desks and behind furniture [6]. In vital-sign sensing, it is common to measure potential signals and obtain electromyograms, such as electrocardiographs [7,8]. However, it is difficult to detect signals such as the electric potential in, for example, a toy poodle, a popular indoor dog, because its body is thickly covered with fur. When measurement is necessary, the fur is shaved in the area where the electrodes are attached, which is a burden on the poodle and distressing for its owner. Even if its accuracy is not yet ideal or high, a system that enables uninterrupted monitoring in combination with web-based surveillance cameras is desired. On the other hand, wearable systems that continuously monitor an individual’s biological profile noninvasively or with minimal invasivity were first realized decades ago, and such systems have progressed rapidly to the present [9,10]. In recent years, there have been many reports on the importance of health monitoring systems, especially those having wearable sensors [11,12,13]. There have also been reports on the practical use of these systems such as the detection of health hazards in the COVID-19 pandemic [14,15]. Pioneering reports of efficient systems using machine learning for data processing have also been published [16,17]. On the other hand, piezoelectric sensors are widely used in harvesting and other applications based on vibration detection [9,18,19,20]. Recently, we developed a braided cord sensor made of piezoelectric poly-*l*-lactic acid (PLLA) fibers (piezoelectric PLLA braided cord). A piezoelectric PLLA braided cord is a unique sensor that can be tied, untied, and even embroidered [21,22]. This makes it a promising material for e-textiles, which are currently the focus of much attention [10,23]. In addition, there are two factors making PLLA an environmentally friendly material [24]. The most important is that it does not contain lead, which causes environmental pollution [25]. This is especially a priority in the European Union (EU), where lead zirconate titanate (PZT) is the most widely used piezoelectric material [26] and researchers in EU countries are searching for alternative lead-free materials. Furthermore, in recent years, the deterioration of the global environment, typhoons, droughts, and the disappearance of icebergs in the Arctic Ocean have been attributed to global warming, and carbon dioxide emissions are being controlled with the goal of preventing further deterioration. For this reason, plant-derived polymers are attracting attention [24,25]. From these perspectives, PLLA is expected to be a highly promising piezoelectric material. However, PLLA has less than 1/100th of the piezoelectricity of PZT and less than 1/20th of that of polyvinylidene fluoride (PVDF), a petroleum polymer. For this reason, most research on PLLA has focused on improving its piezoelectricity [24,25,26]. We have also improved the piezoelectricity of PLLA fibers through many years of research, and we have developed unique sensors and piezoelectric PLLA braided cords [27,28,29,30].

In this study, on the basis of the results of previous studies, a prototype sensing system with a piezoelectric PLLA braided cord was fabricated, which is useful for monitoring the daily movements of indoor dogs and maintaining their health. We report on our findings.

## 2. Improved Piezoelectric Properties While Maintaining High Mechanical Durability

The conditions under which the piezoelectric PLLA braided cord are applied to indoor dogs have significant differences from those for human wearable devices. The conditions are as follows. Indoor dogs make sudden unpredictable and violent movements. For this reason, the piezoelectric PLLA braided cord must be more durable and robust than that used on humans [30]. In addition, indoor dogs may spill food when eating, and they may also become wet. Therefore, piezoelectric PLLA braided cords used as indoor dog sensors must be robust against these unpredictable environmental conditions. Furthermore, piezoelectric response signals generated by the piezoelectric PLLA braided cord may weaken owing to the effect of the fur of indoor dogs. Therefore, it is necessary to increase the piezoelectricity of piezoelectric PLLA braided cords from the current level. The piezoelectricity of PLLA fibers used in the piezoelectric PLLA braided cords developed to date has been improved to the highest extent to enhance their sensing function. As reported previously [31,32,33], we increased the degree of crystallinity and the coefficient of fiber orientation, and controlled the higher-order structure so as to exclude amorphous portions to the greatest extent possible. As a result, current piezoelectric PLLA fibers lose their inherent strength and flexibility, and become brittle. In fact, in a preliminary experiment, the piezoelectric PLLA fiber used in the piezoelectric PLLA braided cord sensor occasionally cracked owing to its brittleness when it was applied to an indoor dog. Therefore, it was necessary to increase the physical durability of the sensor for it to be practically useful. To realize this, on the basis of research results obtained to date, we began by examining the conditions for fabricating fibers with both high piezoelectric constants and mechanical durability.

For use as a sensor for indoor dogs, priority was given to achieving durability to withstand the movements of indoor dogs, rather than to maximizing the piezoelectricity of the PLLA fiber. Since piezoelectricity is derived from the crystalline part, the amorphous region must be reduced in order to improve the piezoelectricity of the entire macroscopic system [34]. In other words, to improve piezoelectricity, it is necessary to increase the degree of crystallinity during stretching. Conventional methods of developing piezoelectricity reduce amorphous regions, resulting in loss of flexibility, suppleness, and elasticity, and, thus, mechanical durability. Therefore, we attempted the following method to achieve both high piezoelectric performance and durability. First, we tried to increase the degree of orientation by increasing the winding speed at the spinning stage. We expected that this would result in the development of high piezoelectricity. Then, by suppressing the elongation during drawing, the decrease in the amorphous region was prevented. In other words, by adjusting the orientation during spinning, we tried to improve flexibility while maintaining piezoelectricity. Many experiments were conducted as a result of trial and error. The result was a piezoelectric PLLA fiber that was effective as a sensor for indoor dogs. Typical measured piezoelectric properties of six PLLA fibers are summarized in Table 1. 

Tensile strength was obtained by converting stress–strain curves obtained on a general-purpose tensile testing machine (Imada FSA 0.5K2) using 10 bundles of PLLA fibers as test specimens. By increasing the spinning speed, we were able to develop a PLLA yarn with lower piezoelectricity than before, while still achieving a piezoelectric constant of about 8 pC/N with 20% stretching, allowing its use as a sensor having supple mechanical properties. A typical PLLA fiber is shown in Figure 1. The higher-order structure was observed by atomic force microscopy (AFM). The results are summarized in Figure 2. The figure shows images of a sample prepared by simple stretching, having piezoelectricity too low for use as a sensor (a), a sample with high piezoelectricity, which was obtained using a previously reported starching process (b) [22,27,30,31,32,33], and a sample that we developed with practical piezoelectricity and improved mechanical properties (c). We can clearly see that a well-developed higher-order structure was necessary to realize a high piezoelectric constant. In contrast, the sample obtained in this experiment had a uniformly developed dense higher-order structure, which was considered to be responsible for the improved mechanical and piezoelectric properties. According to previous reports [35,36,37], as shown in Figure 3, piezoelectric PLLA braided cords are made by bundling and braiding piezoelectric PLLA fibers. The core of piezoelectric PLLA braided cord is a conductive fiber bundle, around which piezoelectric PLLA fibers and water-repellent yarn are wound. Furthermore, the conductive fibers are covered with PLLA fibers and water-repellent yarn to achieve a coaxial cable structure. Previously, insulating PET fibers were used instead of water-repellent yarns, but this time, the insulating PET yarns were changed to water-repellent yarns to improve water resistance for use as a sensor for dogs. Figure 4 shows the results of experiments in which the piezoelectric PLLA braided cord was subjected to bending vibration stress, during which water was applied to the cord. The response signal of the conventional product disappeared almost immediately, but the signal continued to be generated when the water-repellent yarn was used. Sensor 1 is the new piezoelectric braided cord sensor with the water-repellent yarn, and Sensor 2 is the conventional sensor. Sensors 1 and 2 were vibrated at 1 Hz and water droplets were simultaneously applied. Sensor 2 stopped responding almost immediately, but Sensor 1 was unaffected. The experimental results showed that Sensor 1 had improved water resistance.

## 3. Finite Element Method (FEM)

We designed a sensor for indoor dogs using the piezoelectric PLLA braided cords developed in this study with improved mechanical durability. For indoor dogs, such as toy poodles, it is very difficult to perform measurements using piezoelectric PLLA braided cords on a fabric placed on the body surface because the body surface is covered with fur. However, our previous studies showed that piezoelectric PLLA braided cords embroidered on a fabric could measure only stretching or shearing motion, or specific motions, depending on the stitch type [38,39,40]. These results indicated that, depending on the stitch type of the embroidered piezoelectric PLLA braided cords, it was possible to selectively detect the necessary signals from the poodle’s complex movements. Therefore, to design embroidering with the optimal stitching detecting the motion of indoor dogs, FEM analysis was performed using FEMTET software developed by Murata Manufacturing Co. On the basis of the results of this analysis, a prototype poodle sensor was fabricated. First, to clarify the piezoelectric properties of the piezoelectric PLLA braided cord with increased mechanical strength and water repellency, calculations were carried out for a model in which the piezoelectric PLLA braided cord was sewn in a straight line into a fabric. When an extension strain of 1% was applied to the fabric, an electric field was generated at the intersection of the piezoelectric PLLA braided cord and the fabric owing to the piezoelectric response. The green circles in Figure 5 indicate where the piezoelectric response was generated. As with the conventional product, a piezoelectric signal was confirmed at the point of intersection. As can be seen from Figure 5, an electric field of observable magnitude was generated where the piezoelectric PLLA braided cord was pre-bent to form a nodal point. This result was very important. For example, suppose that a piezoelectric PLLA braided cord is embroidered on a fabric. Its piezoelectric response is dependent on the stitch type of the embroidery. In other words, the piezoelectric response signal from the embroidered piezoelectric PLLA braided cord is obtained only when the appropriate stitch type is selected such that the pre-bent piezoelectric braided cord is effectively positioned for the applied displacement or stress, i.e., only the signals necessary for displacement are selectively generated. This is the greatest advantage of our newly developed piezoelectric PLLA braided cord, which cannot be realized with other sensor materials that use piezoelectric PLLA braided cords as embroidery sensors.

From these results, we calculated samples of piezoelectric braided cords intertwined with each other as they are in embroidery. Specifically, piezoelectric PLLA braided cords connected in the form of a loop, as shown in Figure 6. The piezoelectric response occurs at the intersection points, respectively. From these results, the basic shape of the embroidery, a loop, was continuously connected and analyzed. Specifically, we calculated the piezoelectric response to stresses applied parallel and perpendicular to the direction of the array of loops. It can be seen in Figure 7 that the piezoelectric response depended on whether the loops were perpendicular or parallel. The results of this analysis were then used to examine the effect type of the stitch. The results confirmed that applying the basic stitch types effectively generated a piezoelectric response when the loops were parallel to the fabric. Therefore, we examined the effectiveness of different types of stitch to be sewn on clothing. Specifically, of the five basic types of embroidery stitch (cross, feather, chain, backstitch, and running stitch), we examined chain, feather, and cross stitches, in which the loops lie on the fabric, and running stitches, in which the number of crossing points can be easily adjusted. The results are shown in Figure 8 and reveal that the chain stitch had the highest piezoelectric response sensitivity. It was also found that, as the density (number of intersections) of the loops increased, the response sensitivity to minute displacements increased. These results confirmed that using stitches with a basic shape effectively generated a piezoelectric response when the loops were parallel to the fabric.

## 4. Design of Sensor for Indoor Dogs Using Piezoelectric Braid

### 4.1. Dog Wear for Toy Poodles Embroidered with Piezoelectric PLLA Braided Cords

On the basis of the aforementioned FEM results, the chain stitch was selected for embroidering the piezoelectric PLLA braided cord for the prototype sensor. A piezoelectric PLLA braided cord was placed as a pulsation sensor in the embroidery around the neck, where the vibration was smallest and would not be disturbed by body movement. Another piezoelectric PLLA braided cord was placed as a respiration sensor around the abdomen, since indoor dogs mainly breathe through their abdomen, and was placed in such a way that it was not affected by leg movements. Furthermore, another piezoelectric PLLA braided cord was placed as a locomotion sensor on the hind limbs because of their large movements during jumping. Figure 9 shows dog wear for a poodle in which the piezoelectric PLLA braided cord, with improved mechanical properties and water resistance developed in this study, was embroidered in each of the above-mentioned locations using normal embroidery techniques. The fabric of the dog wear (the red part) is elastic so that the dog wear wraps around the body. Finally, the dog wear was tailored so that it does not unduly hinder the movements of the poodle or cause it discomfort.

### 4.2. Sensing Functionality of Dog Wear Embroidered with Piezoelectric PLLA Braided Cords

For reference, we prepared an electrocardiograph potential measurement-type sensor, a pet harness-type sensor manufactured by Sharp Corporation with high accuracy in ECG measurements at rest (details of this sensor: https://corporate.jp.sharp/news/180611-b.html) (accessed on 20 September 2020). To attach the electrodes of this pet harness-type sensor, the poodle’s body hair was partially shaved, and a conductive gel was applied for complete attachment. The dog was then dressed in the dog wear embroidered with the piezoelectric PLLA braided cord over the electrodes of pet harness-type sensor. In the measurement to determine the accuracy of the piezoelectric PLLA braided cord embroidered on the dog wear, the dog was kept at rest to ensure the accuracy of the electrocardiograph-type sensor, which performed measurements simultaneously and was used as a reference. Simultaneously, the piezoelectric braided cord embroidered around the neck of the dog wear was used to detect pulsation. As shown in Figure 10, the response signals obtained from the piezoelectric PLLA braided cord, and those from the electrocardiograph-type sensor, had slightly different waveforms, such as the presence of noise, because the piezoelectric PLLA braided cord sensed pulsation and the electrocardiograph-type sensor measured the change in potential. Therefore, the peak times indicated by each signal were also slightly different. However, the number of peaks per unit time, the so-called pulse rate, obtained from the piezoelectric PLLA braided cord was in good agreement with that obtained from the electrocardiograph-type sensor. According to veterinary books, the normal pulse rate is approximately 60 to 80 beats per minute for small dogs and 40 to 50 beats per minute for large dogs [7,8]. This suggested that the pulse rate obtained in this experiment was reliable. The results confirmed that when the dog was at rest, the piezoelectric PLLA braided cord embroidered around the neck of the dog wear sensed pulsations as designed, and its accuracy was equivalent to that of commercially available electrocardiograph-type sensors.

Next, the sensing function of the piezoelectric braided cord embroidered under the right armpit of the dog wear to measure respiration was examined. At the same time, a commercially available abdominal belt respirometer was worn. The commercially available abdominal belt respirometer used here was not capable of measuring waveforms but clearly indicated the number of breaths per minute. The results are shown in Figure 11, together with the response signal obtained from the piezoelectric braided cord. Assuming that the peak of the response signal from the piezoelectric braided cord corresponded to the respiratory rate, the average peak interval was 23.7 breaths per minute. In contrast, the value obtained using the commercial measuring device was 24.2 breaths per minute [7,8]. These results were in good agreement, indicating that the piezoelectric braided cord embroidered on the right abdomen of the dog wear could accurately measure the respiratory rate during motion. The results obtained were considered valid. The values for vital signs obtained in the above experiments were consistent with those reported in veterinary books. Thus, it could be concluded that the piezoelectric braided cord embroidered on the neck and on the right side of the dog wear, to respectively sense pulsation and respiration, achieved the desired accuracy.

After determining that pulsation and respiration could be detected, we investigated whether the remaining piezoelectric braided cords embroidered on the shoulder and of the dog wear could detect the poodle’s movement during exercise. As shown in Figure 12, no periodic waves corresponding to respiration and pulsation were detected by the sensors in these two locations. On the other hand, we observed clear piezoelectric signals only when the dog was moving. Furthermore, the results obtained when the dog was standing reasonably still suggested that the sensors also detected small movements specific to dogs, as described in veterinary books.

## 5. Prototype Demonstration of an Indoor Dog Monitoring System Using Piezoelectric PLLA Braided Cords Embroidered on Dog Wear as Sensors

The basic experiment described in the previous section revealed that the piezoelectric PLLA braided cord could be used for sensing vital signs, such as pulse and respiration, and motion sensing, by embroidering the piezoelectric PLLA braided cord at designated points on the dog wear using chain stitch, as evidenced in the results of the finite element method study. Again, for sensors in an indoor dog monitoring system to be wearable, it is strongly required that the indoor dog is not burdened by wearing such a system and that the dog’s movement is not restricted in daily life. From this premise, the purpose of the experiment on the prototype system was to take advantage of the selectivity of the piezoelectric braided cord for motion sensing and to show that the main movements of an indoor dog could be determined in a very simple way without constructing a complicated decision algorithm. For example, it is technically possible to build a complex circuit system combined with a PC that can accurately determine the activity of an indoor dog. However, it is impossible to make the system lightweight and compact enough to not limit the activity of the dog; this is considering only the weight of a mobile battery. In other words, the data acquisition and transmission circuits must be as simple as possible and could be driven by a coin-operated power supply. On the basis of these requirements for practical use, we determined the basic behavior of an indoor dog without complex processing of the sensing signals from piezoelectric braided cords embroidered in four locations on the dog’s clothing. Specifically, we attempted to classify resting, walking, standing up, and jumping (to please the owner) using our protype system, and determined the accuracy of the system in classifying these movements.

### 5.1. Data Acquisition and Transmission Systems

A small and lightweight circuit system that does not interfere with the movement of indoor dogs is needed. Therefore, we developed a 2 cm square circuit, as shown in Figure 13. To achieve this small size and light weight, we used an ultralow-power Silicon on Thin Buried Oxide (SOTB) microcontroller. Developed by Renesas, for data acquisition and the latest 0.2 mm thin Li battery, manufactured by Nippon Insulator Co., for supply. This battery can currently be used for one day’s worth of experiments on a single charge. Therefore, the circuit is of a size and weight that does not cause discomfort to indoor dogs. The circuit system is briefly described below. The piezoelectric braided cord has a high impedance because it is a polymer dielectric with piezoelectric properties. The input impedance of the analog circuit for detection must also be high. The analog circuit has an operational amplifier for impedance matching, a filter circuit for disturbance noise removal, and an amplification circuit. In practice, a response signal generated by the piezoelectric PLLA braided cord is sent to a preamplifier for impedance conversion. Then, a bandpass filter passes the analog signal with a frequency of 0.3 to 10 Hz, and the passed signal is amplified 200 times by the amplifier. This analog signal is input to the Analog to Digital (A/D) converter circuit on the SOTB and processed by the microcontroller circuit. This single module circuit can process signals from two piezoelectric braided cords simultaneously. The microcontroller performs A/D conversion of the analog data from the two channels with a signal strength of 12 bits and a sampling frequency of 125 Hz. The acquired data are sent to a smartphone via the Bluetooth Low Energy (BLE) communication system. A RF connector ((MHF4L:I-PEX) is swaged on the end of the piezoelectric PLLA braided cord using the connector-making method for coaxial cables.

### 5.2. Concept of Activity Level Determination

As summarized in the previous section, the piezoelectric PLLA braided cord embroidered on the toy poodle’s clothing showed the possibility of sensing the dog’s movement without placing a burden on the toy poodle by using our prototype’s small and light circuit. However, if a simple algorithm cannot detect the poodle’s behavior, a high-precision PC would still be required, and the system would become too large, placing a heavy burden on the poodle and limiting its movement, making the system unusable in practice. The purpose of the experiment on the prototype system presented here is to demonstrate that the selectivity of the piezoelectric PLLA braided cord for motion sensing, depending on the type of embroidery stitch, could be used to determine the primary motion of an indoor dog in a very simple way without the need to construct a complicated decision algorithm. If this possibility could be demonstrated, it opens the way to the construction of a very simple system, eliminating the need for a PC, which requires complex algorithm analysis. As a first step, we decided to use the following simple classification of movements in this study:

(1) Resting (motionless). When there is no motion, hardly any channel emits any signal from piezoelectric PLLA braided cords.

(2) Standing (standing still). Only the piezoelectric PLLA braided cord embroidered on the neck part of the dog wear produces a large signal, whereas the other piezoelectric PLLA braided cords do not produce much signal.

(3) Walking (moving). All piezoelectric PLLA braided cords emit signals of some magnitude.

(4) Jumping (to please the owner). Jumping and wriggling produces a large signal from the piezoelectric PLLA braided cord embroidered on the belly part of the dog wear.

Again, the purpose of this experiment was to demonstrate the extent to which it was possible to determine the movement of an animal using only the signals obtained from these four embroidered piezoelectric PLLA braided cords, without processing the signal output with Fourier transforms or correlation coefficients. If correct judgment could not be demonstrated, it would be complicated because a PC for complex algorithm analysis would be required. If it could be demonstrated, it would eliminates such a requirement, and pave the way for the development of a very simple system. The judgment for the combinations determined here is shown in Table 2. The actual judgment flow is briefly described below. Here, signals from each piezoelectric PLLA braided cord were judged individually, and a decision of “1” was made when a signal was detected. For example, if a large signal was detected from the piezoelectric braided cord embroidered on the neck or flank and almost no signal was detected from the piezoelectric braided cord embroidered on the shoulder or belly, the judgment result was “1010”. This was repeated for the length of the data string and transmitted to a smartphone via BLE wireless communication to display the judgment result of “walking”. This experimental system was very simple and took advantage of the motion selectivity feature of the piezoelectric braided cord, making it a challenging configuration that could not be achieved with conventional sensors.

### 5.3. Verification Experimen

A toy poodle wearing a dog outfit embroidered with the piezoelectric PLLA braided cord was allowed to spend 5 h of free time with its owner in a 10 m × 4 m conference room, as shown in Figure 14. The results obtained using the prototype system were transmitted to a smartphone via BLE communication. The obtained activity status judgment results were expressed by our own software. The results obtained in this way were compared with video images captured by a video camera to determine their authenticity. Figure 15, Figure 16, Figure 17 and Figure 18 are representative examples of cases where the judgment result and the video image matched. Here, the left and right images in Figure 17 are the results for different dogs. The above methods were used to verify the authenticity of the experimental results obtained in this study.

Table 3 shows that the agreement rates during the experiment were 85% when the dog was resting, 75% when it was standing still, 90% when it was walking, and 95% when it was jumping. The main cases of misrecognition were as follows. When it was judged to be walking in Figure 19, the dog actually stopped. After 20 s the dog was judged to have stopped, which was the correct answer. This misjudgment could be attributed to the body movement signals generated when the dog set down and moved its legs at this time. In another case, the dog jumped in Figure 16, which was correct, but then continued to play for 20 s later, during which time the dog was misjudged to be walking because the neck signal became larger and the belly signal became smaller. Although misjudgments do occur as described above, it is surprising that a simple method, such as the one shown in the table, showed a considerable agreement rate.

Finally, we considered the remaining issues in this study. Analysis of the signals from the embroidered piezoelectric PLLA braided cord could provide respiration and pulse rates with accuracy comparable to those determined using commercially available electrocardiograph-type devices. A typical example is shown in Figure 19. The figure shows the pulse and respiration rates of a poodle when it was standing still, resting, and jumping. Unfortunately, the accuracy of the signals from the piezoelectric PLLA braided cord could not be verified because there was no device that could accurately measure the pulse rate of the poodle during exercise without applying a load to the poodle. Furthermore, in human medical practice, the degree of stress is commonly measured from changes in pulse rate and expressed as an LF/HF index [41,42,43,44]. It is also used as an indicator of sleep quality and workload. This is shown in Figure 19c. Unlike in humans, its accuracy for dogs cannot be confirmed by a questionnaire method, and its use required guidance by a veterinary expert. Pioneering research into methods for humans to remotely interact with their pets using IoT devices has been actively pursued and is producing significant results. To contribute to the development of this field [45,46], we plan to collaborate with researchers and veterinarians in this field to make LF/HF an objective indicator. Since the use of LF/HF is assumed to be closely related to the daily mental state and health of indoor dogs, we would like to actively seek guidance by a veterinary expert and explore ways to make effective use of LF/HF.

In the above experiments, we used only the piezoelectric signals from the four piezoelectric PLLA braided cords embroidered on the dog wear to discriminate basic dog movements without complicated algorithm processing and found that the discriminations could be made with considerable accuracy. This is an epoch-making achievement based on the selective motion sensitivity of the piezoelectric PLLA braided cord. In other words, the simple sensing system described here is unprecedented, compact, and lightweight, and it does not place a heavy burden on the poodle. This achievement has paved the way for its practical application. However, there is room for improvement in the mounting position of the sensors and the accuracy of the algorithm. In addition, the discrimination algorithm could be improved by using data alignment and standard deviation in the program to reduce misjudgments. We plan to further improve the algorithm in the future.

## 6. Conclusions

We examined the potential of the sensing functionality of piezoelectric PLLA braided cords to realize a better indoor dog-monitoring system that compensates for the weak points of webcam-based indoor dog-watching systems, namely, camera vision, and acquires detailed information on the dog’s activity status and vital signs. The PLLA fibers used in the piezoelectric PLLA braided cord were spun in a way that maintained their mechanical durability against dog movements, and their higher-order structure was improved. In addition, since indoor dogs, such as toy poodles, are furry, and sensing using the piezoelectric PLLA braided cord is difficult, a method of embroidering the piezoelectric PLLA braided cord by FEM analysis was investigated. As a result, it was found that among various embroidery stitches, the chain stitch could achieve selectivity of movement sensing by the piezoelectric PLLA braided cord. A prototype system was constructed by embroidering piezoelectric braided cords on the dog wear and fabricating data processing and transmission circuits. The prototype system was actually worn by a poodle, and the activity was determined using a simple algorithm that did not involve complicated algorithm processing. The experimental results obtained in this study strongly suggest the possibility of an innovative monitoring system that does not place a burden on small indoor dogs. Although the sensing function of the prototype system reported here was satisfactory, durability tests of PLLA fibers necessary for practical use are also required. In order to realize piezoelectric PLLA fibers in practical use as wearable sensors, the following weaknesses must be considered.

Weaknesses: Water vapor

X Put in washing machine or dryer.

X Stiffen with starch.

X Steam ironing.

X Use detergent (pH neutral detergent is OK.)

For practical use, it is necessary to propose a method of use that takes the above weaknesses into account. Full-scale durability tests of the prototype system reported here under various environments will be conducted in the future.

## Figures and Tables

**Figure 1 micromachines-14-00143-f001:**
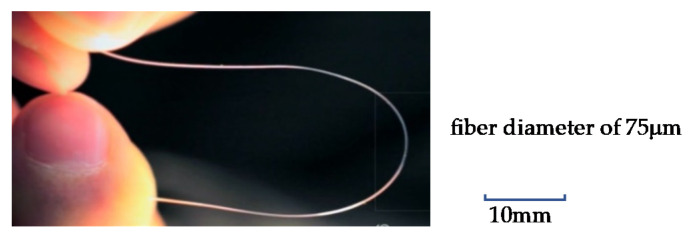
PLLA fiber with practical piezoelectric properties and flexibility for sensing.

**Figure 2 micromachines-14-00143-f002:**
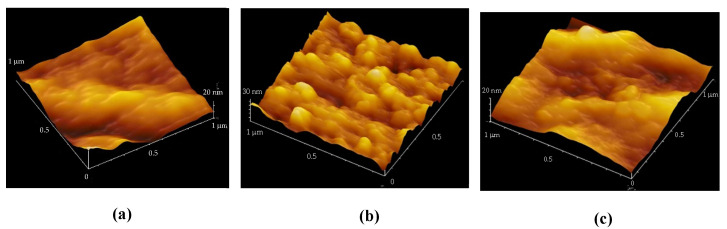
AFM images. (**a**) Sample with low piezoelectricity. (**b**) Sample with poor mechanical properties and high piezoelectricity (previously reported). (**c**) Developed PLLA fiber with good mechanical and piezoelectric properties for sensing.

**Figure 3 micromachines-14-00143-f003:**
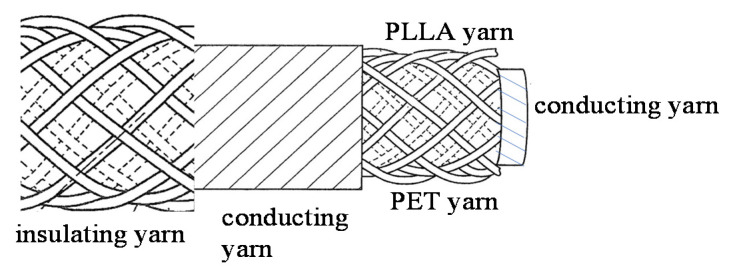
Piezoelectric PLLA braided cord.

**Figure 4 micromachines-14-00143-f004:**
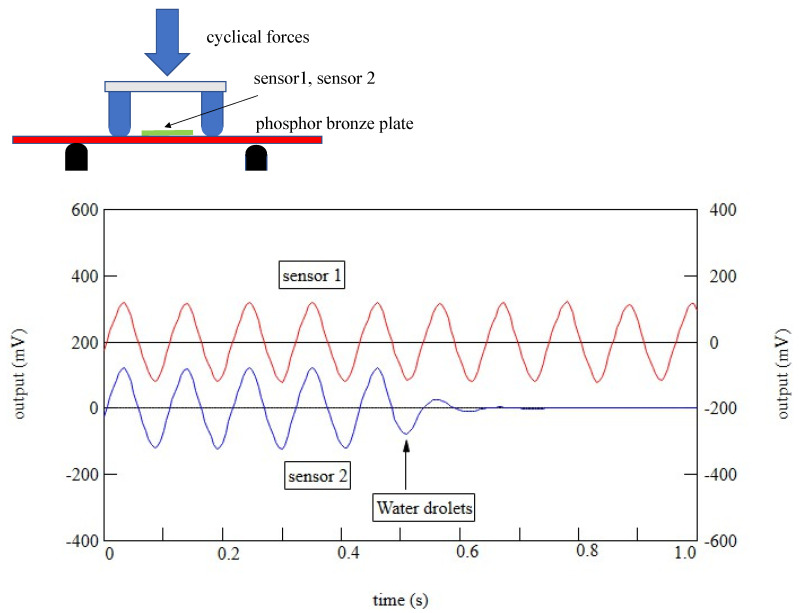
Piezoelectric response signals of conventional and water-resistant piezoelectric braided cords under water droplets.

**Figure 5 micromachines-14-00143-f005:**
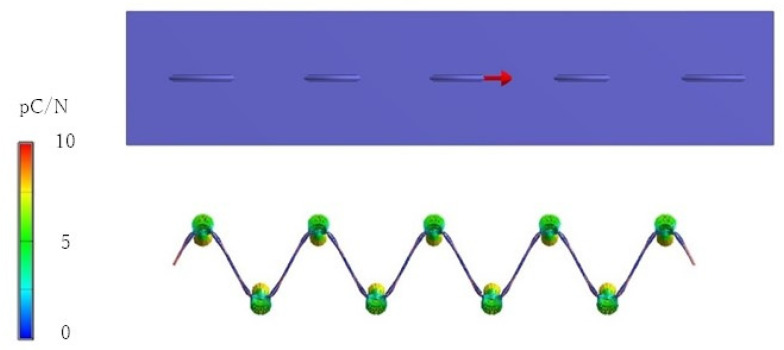
FEM calculation of piezoelectric response of a single piezoelectric PLLA braided cord sewn into the fabric under applied strain to the fabric (upper figure: top view, lower figure: cross-sectional view).

**Figure 6 micromachines-14-00143-f006:**
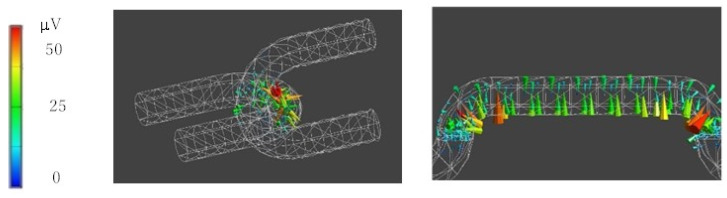
Piezoelectric response of an entangled piezoelectric braided cord under tensile stress.

**Figure 7 micromachines-14-00143-f007:**
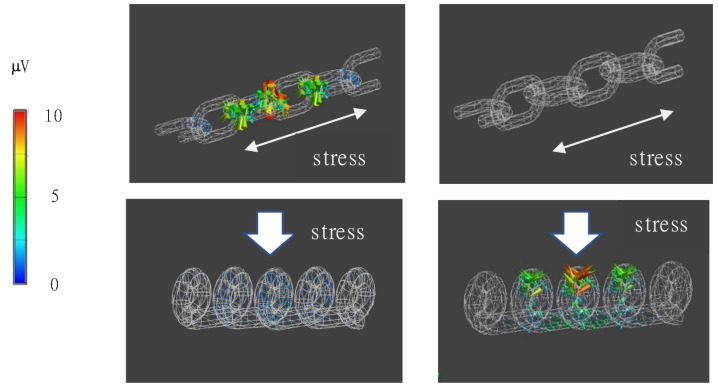
Piezoelectric response of piezoelectric braided cord that mimics embroidery stitches.

**Figure 8 micromachines-14-00143-f008:**
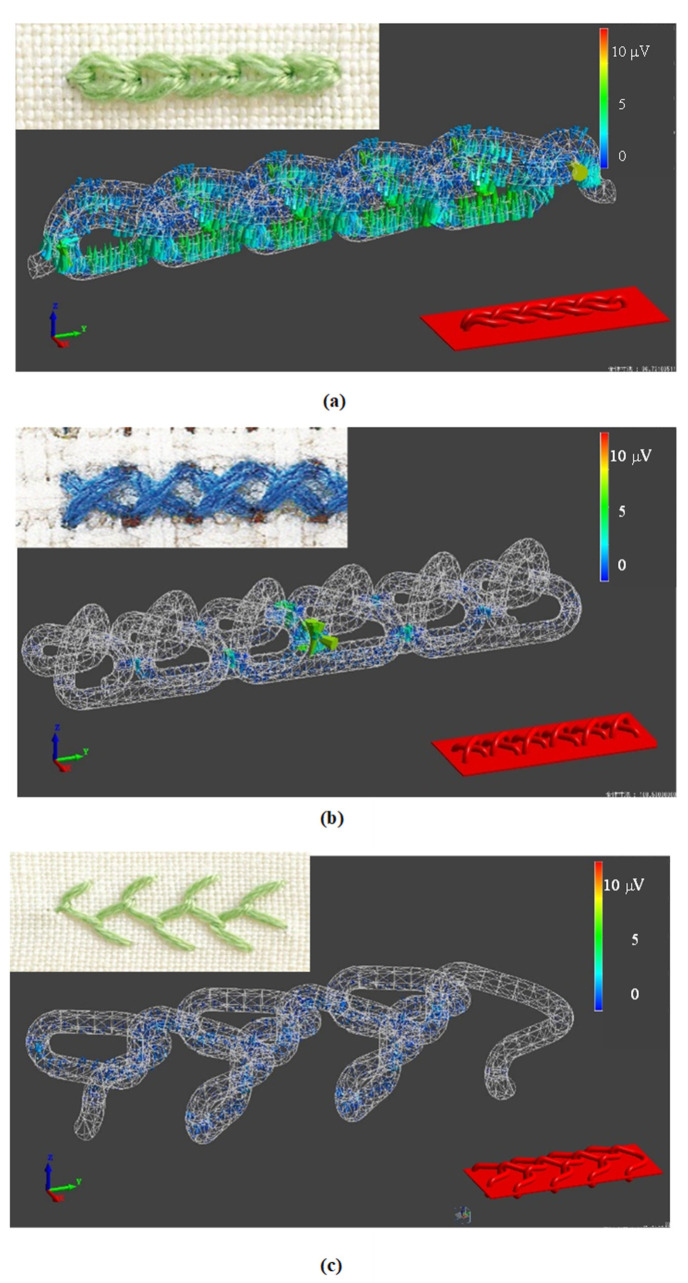
Piezoelectric response of piezoelectric braided cord embroidered using chain stitch (**a**), cross stitch (**b**), and feather stitch (**c**).

**Figure 9 micromachines-14-00143-f009:**
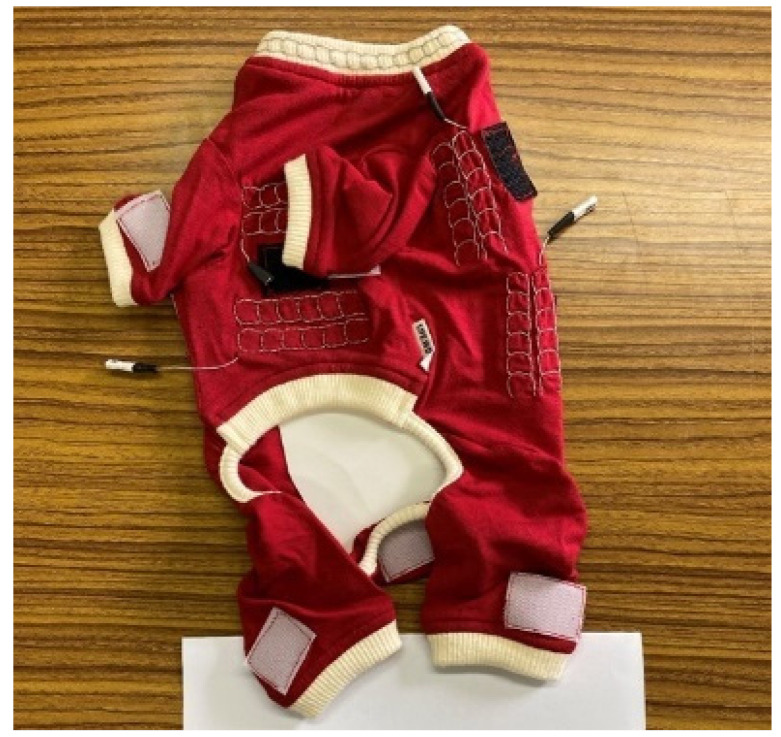
Dog wear for toy poodles embroidered with piezoelectric PLLA braided cords using chain stitch.

**Figure 10 micromachines-14-00143-f010:**
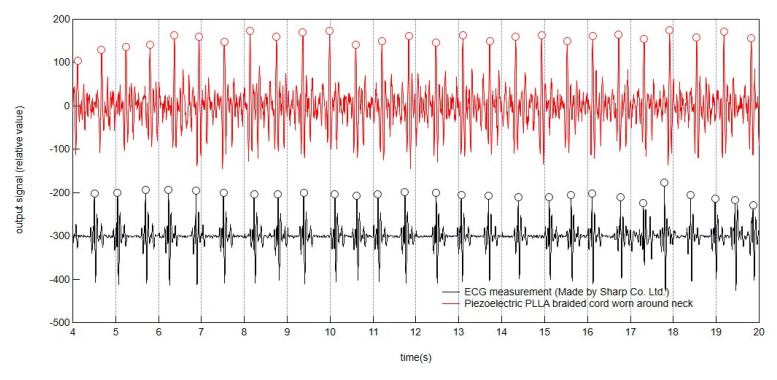
Waveforms of response signals from a piezoelectric PLLA braided cord embroidered around the neck of dog wear and a commercial electrocardiograph-type sensor for poodle at rest.

**Figure 11 micromachines-14-00143-f011:**
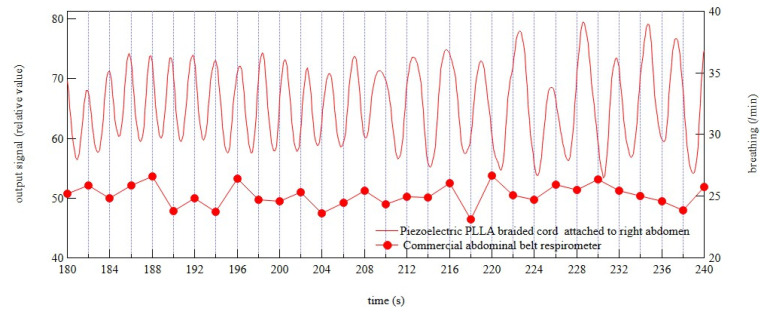
Response signal from a sensor embroidered on the right abdomen of the dog wear and respiratory rate from a commercial abdominal belt respirometer. The dog was measured while being held by its owner.

**Figure 12 micromachines-14-00143-f012:**
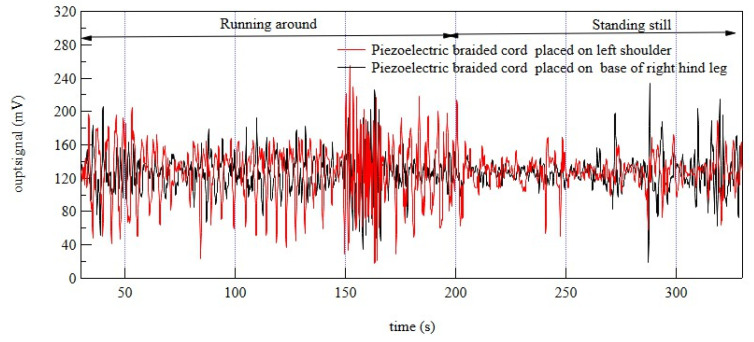
Response signals from piezoelectric PLLA braided cords embroidered on the shoulders and base of hind legs of the dog wear when the toy poodle is in motion and at rest.

**Figure 13 micromachines-14-00143-f013:**
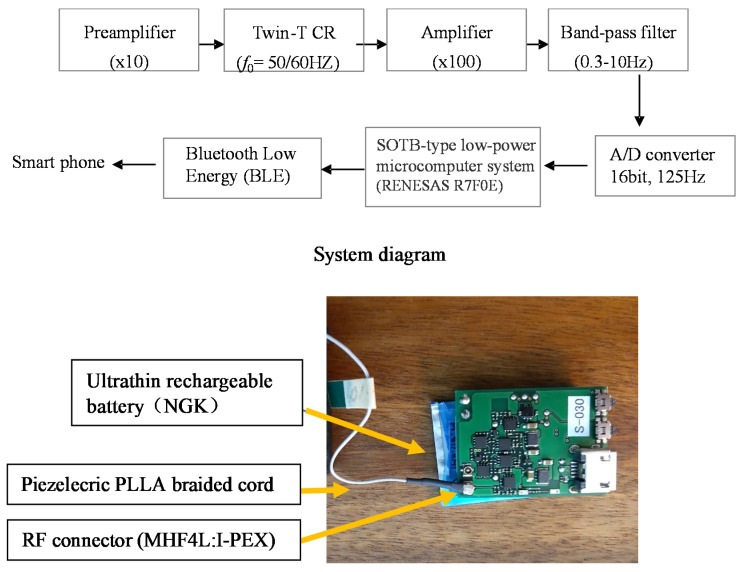
Circuit system diagram for piezoelectric PLLA braided cord.

**Figure 14 micromachines-14-00143-f014:**
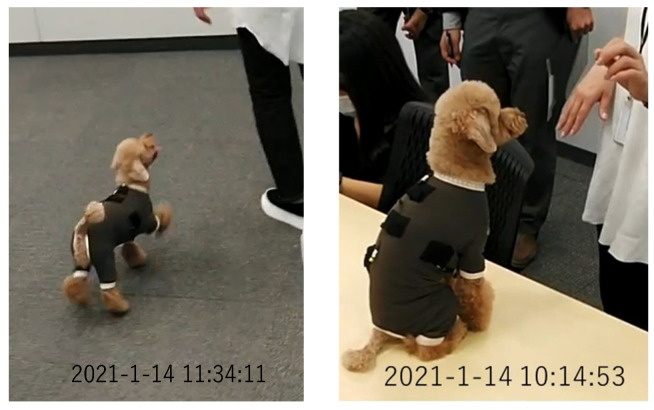
Clip of video footage of a toy poodle taken with a video camera installed in a conference room during the experiment.

**Figure 15 micromachines-14-00143-f015:**
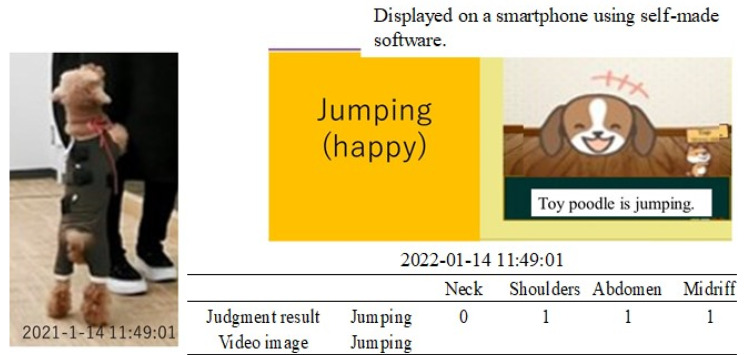
Example of match between the judgment result and the video image.

**Figure 16 micromachines-14-00143-f016:**
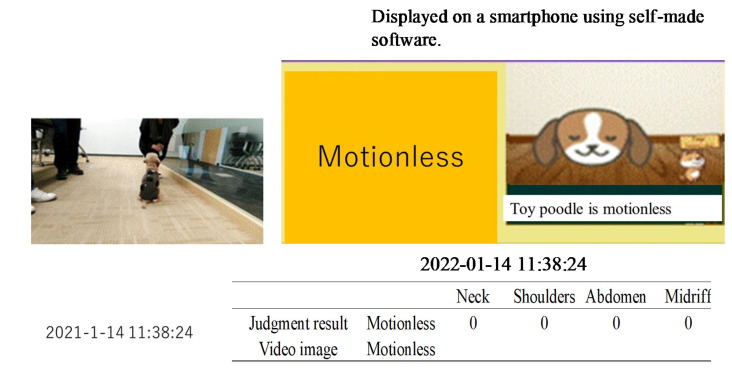
Example of match between the judgment result and the video image.

**Figure 17 micromachines-14-00143-f017:**
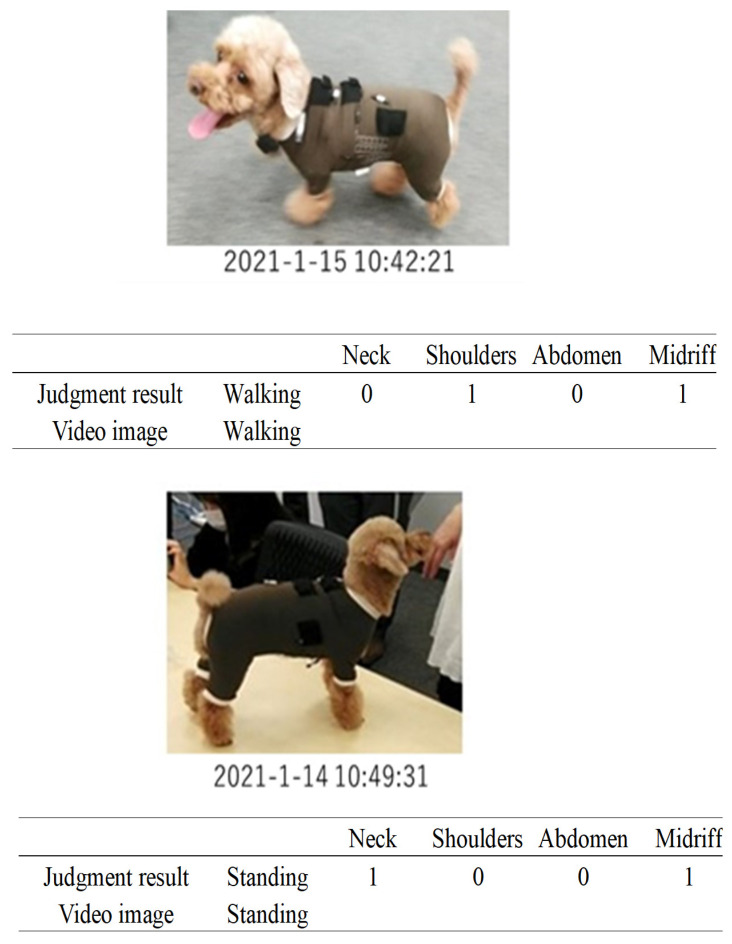
Examples of various matches between the judgment results and the video image.

**Figure 18 micromachines-14-00143-f018:**
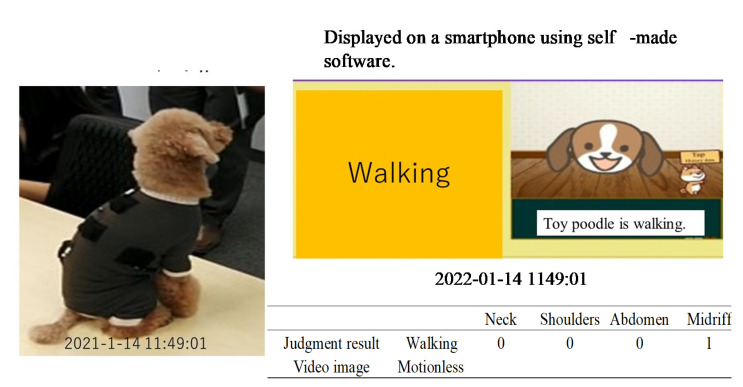
Example of inconsistency between the judgment result and the video image.

**Figure 19 micromachines-14-00143-f019:**
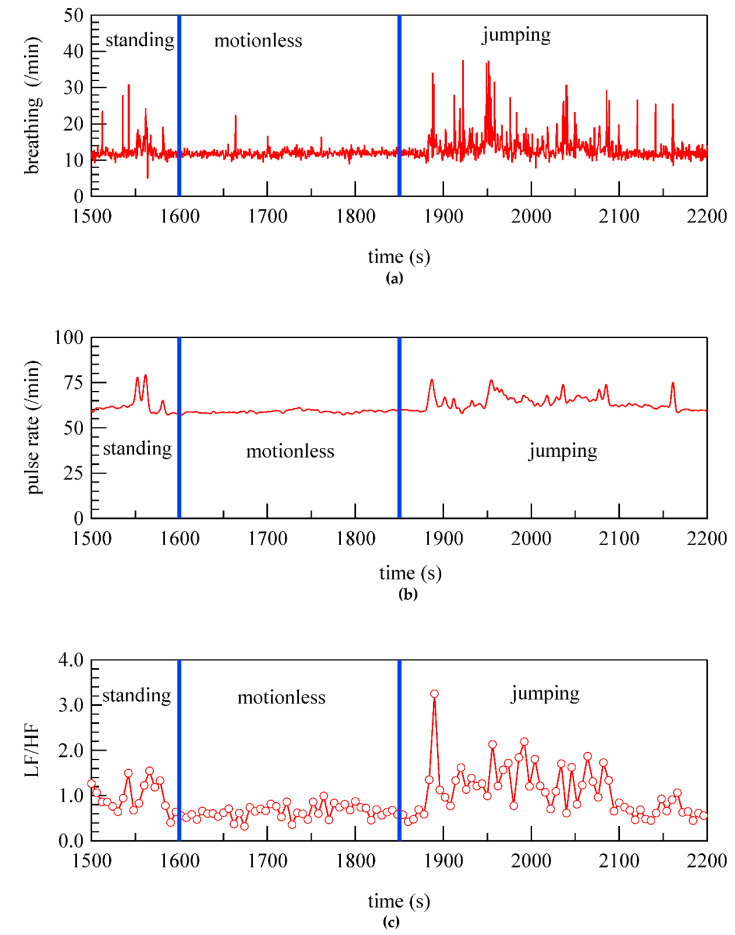
Examples of vital signs obtained from piezoelectric PLLA braided cord: (**a**) respiratory rate (**b**) pulse rate (**c**) LF/HF.

**Table 1 micromachines-14-00143-t001:** Physical properties of PLLA fibers.

Molecular weight (×10^3^)	200	200	200	240	240	240
Spinning speed (m/min)	800	800	800	1600	1600	1600
Drawing ratio and temperature	×10 (97 °C)	×6 (100 °C)	×4 (100 °C)	×3.5 (95 °C)	×2.5 (95 °C)	×2 (95 °C)
Melting point (°C)	184.3	183.8	184.2	184.2	184.7	181.7
Degree of crystallinity (%)	58.1	51.5	51.5	52.5	53.5	50.6
Coefficient of orientation	0.986	0.912	0.788	0.867	0.706	0.701
Stretching (%)	8–10	15–20	25–30	20–30	25–30	28–32
Tensile strength (GPa)	0.20	0.30	0.35	0.55	0.54	0.51
Piezoelectric constant (pC/N)	20.1	17.5	6.0	8.2	4.8	5.0

**Table 2 micromachines-14-00143-t002:** Decision table.

Threshold Judgment				State of Activity
Neck	Shoulders	Abdomen	Midriff	
0	0	0	0	motionless
0	0	0	1	walking
0	0	1	0	walking
0	0	1	1	walking
0	1	0	0	walking
0	1	0	1	walking
0	1	1	0	walking
0	1	1	1	jumping (happy)
1	0	0	0	standing
1	0	0	1	standing
1	0	1	0	walking
1	0	1	1	jumping (happy)
1	1	0	0	walking
1	1	0	1	walking
1	1	1	0	walking
1	1	1	1	jumping (happy)

**Table 3 micromachines-14-00143-t003:** Rate of agreement between the judgment result from the signal from the piezoelectric blade and the video image.

	Resting (Motionless)	Standing Still	Walking (Moving)	Jumping (to Please the Owner)
Rate of agreement	85%	75%	90%	95%

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
