# Peer review of "Application of Piezoelectric PLLA Braided Cord as Wearable Sensor to Realize Monitoring System for Indoor Dogs with Less Physical or Mental Stress"

_micromachines, 2023, doi:10.3390/mi14010143_

Round 1

Reviewer 1 Report

The article presents a development and validation of a braided PLLA sensor array for dog monitoring applications. The work well describes the process from material design and braid configuration to final application including analysis of sensor movement data on dog positions.

Overall the work describes the sensor design and experimental results. The publication can be improved in a few ways:

Line 151: Figure 4 would benefit from an image of the "bending vibration stress" experimental set up.

Line 201: spelling error, "Figure 8 are reveal"

Line 245: spelling error, "each signal are also slightly differ different"

Line 211: in the prototype design, how exactly where the PLLA sensor bundles electrically connected to the sensing circuit? Was a conductive epoxy used or a similar method? Were there issues with creating a stable connection? Would the existing process be able to scale easily for production?

Line 386: Figure 15 - 18 the tables are difficult to read, the font size should be increased and the table background removed. Also the black text on green background should be changed to be easy to read.

Author Response

Dear Reviewer,

Thank you very much for your helpful remarks.

We have revised the manuscript as follows.

>Line 151: Figure 4 would benefit from an image of the "bending vibration stress" experimental set up.

>Line 201: spelling error, "Figure 8 are reveal"

We have corrected it.

>Line 245: Spelling error, "Each signal is also slightly different.

We have corrected.

> Line 211: in the prototype design, how exactly where the PLLA sensor bundles electrically connected to the sensing circuit? Was a conductive epoxy used or a similar method? Were there issues with creating a stable connection? Would the existing process be able to scale easily for production?

Figure 13 shows new photograph of the  end connection of piezoelectric PLLA braided cord (line 324). Since this connection is manufactured using normal industrial methods, the production scale could be easily scaled up.

>Line 386: Figure 15 - 18 the tables are difficult to read, the font size should be increased and the table background removed. Also the black text on green background should be changed to be easy to read.

We are very sorry for the inconvenience caused.

We have rewritten it.

Thank you very much.

Yours truly,

Prof. Y. Tajitsu, Ph.D.

Reviewer 2 Report

This study presents a fabrication method for piezoelectric PLLA braided cords. Considering piezoelectric fiber is drawing a tremendous attention in these days as the wearable electronics became a popular research topic, this paper provides an enough valuable study on the wearable electronics and can be recommended for publication after the following revision comments are reflected in the revision. 

1. In Figure 1, the scale bar should be shown to let readers know the practical size of PLLA fibers. 

2. Any durability test for PLLA fibers, for example, washing, bending, or twisting? 

Author Response

Dear Reviewer,

Thank you very much for your helpful remarks.

We have revised the manuscript as follows.

>1. In Figure 1, the scale bar should be shown to let readers know the practical size of PLLA fibers.

Scale and fiber diameter have been added to the figure.

>2. Any durability test for PLLA fibers, for example, washing, bending, or twisting?

We are very sorry for the insufficient description. Following your suggestion, we have added the results known to date after line 448.

Thank you very much.

Yours truly,

Prof. Y. Tajitsu, Ph.D.
